# Long-Term Phenotypic and Proteomic Changes Following Vitrified Embryo Transfer in the Rabbit Model

**DOI:** 10.3390/ani10061043

**Published:** 2020-06-17

**Authors:** Ximo Garcia-Dominguez, Francisco Marco-Jiménez, David S. Peñaranda, José Salvador Vicente

**Affiliations:** Instituto de Ciencia y Tecnología Animal, Universitat Politècnica de València, 46022 Valencia, Spain; ximo.garciadominguez@gmail.com (X.G.-D.); fmarco@dca.upv.es (F.M.-J.); dasncpea@upvnet.upv.es (D.S.P.)

**Keywords:** assisted reproduction technology, embryo vitrification, embryo transfer, postnatal outcomes, proteome

## Abstract

**Simple Summary:**

This study was conducted to demonstrate how a vitrified embryo transfer procedure incurs phenotypic and molecular changes throughout life. This study reports the first evidence describing that embryonic manipulation during a vitrified embryo transfer cycle induced molecular modifications, concerning oxidative phosphorylation and dysregulations in zinc and lipid metabolism in liver tissue, which has been reported as responsible for postnatal variations of the phenotype.

**Abstract:**

Nowadays, assisted reproductive technologies (ARTs) are considered valuable contributors to our past, but a future without their use is inconceivable. However, in recent years, several studies have evidenced a potential impact of ART on long-term development in mammal species. To date, the long-term follow-up data are still limited. So far, studies have mainly focused on in vitro fertilization or in vitro culture, with information from gametes/embryos cryopreservation field being practically missing. Herein, we report an approach to determine whether a vitrified embryo transfer procedure would have long-term consequences on the offspring. Using the rabbit as a model, we compared animals derived from vitrified-transferred embryos versus those naturally conceived, studying the growth performance, plus the weight throughout life, and the internal organs/tissues phenotype. The healthy status was assessed over the hematological and biochemical parameters in peripheral blood. Additionally, a comparative proteomic analysis was conducted in the liver tissue to investigate molecular cues related to vitrified embryo transfer in an adult tissue. After vitrified embryo transfer, birth weight was increased, and the growth performance was diminished in a sex-specific manner. In addition, vitrified-transferred animals showed significantly lower body, liver and heart weights in adulthood. Molecular analyses revealed that vitrified embryo transfer triggers reprogramming of the liver proteome. Functional analysis of the differentially expressed proteins showed changes in relation to oxidative phosphorylation and dysregulations in the zinc and lipid metabolism, which has been reported as possible causes of a disturbed growth pattern. Therefore, we conclude that vitrified embryo transfer is not a neutral procedure, and it incurs long-term effects in the offspring both at phenotypic and molecular levels. These results described a striking example of the developmental plasticity exhibited by the mammalian embryo.

## 1. Introduction

Since their implementation, assisted reproductive technologies (ARTs) have made major contributions to human health, livestock production and environmental management in the past and will continue to do so in future [1,2]. However, from the outset there has been concern about the influence of these technologies in embryo and postnatal development. This is the basis of the Developmental Origins of Health and Disease (DOHaD) theory, which posits that environmental stresses during development can increase the risk of disease later in life [3]. At the center of the DOHaD theory is the concept of developmental plasticity, which implies that the biological pathways governing prenatal development are not fixed [3,4]. Therefore, in response to environmental signals, this developmental plasticity allows phenotypic changes in developing embryos to be better suited to the environment in which they will be born. Nevertheless, extreme stressors, such as ART conditions, can force developing embryos to carry out a strong reshaping of their developmental trajectories to guarantee their short-term survival [3,4,5]. However, this reprogramming in embryonic cells that subsequently forms tissues and organ systems may influence an individual’s susceptibility to alterations in growth, development and health [3,6,7,8,9,10,11,12,13]. Controversially, ART has been linked in humans with adverse obstetric and perinatal outcomes, birth defects, cancers, growth disorders, obesity and chronic ageing-related diseases [10]. But, in human studies, it is difficult to discriminate whether these effects are caused by ARTs or originate from either genetic abnormalities or risk factors intrinsic to infertile patients with low-quality gametes [9,10]. Thus, animal models using fertile and healthy animals, which avoid these confounding factors and provide adequate control groups, are essential to reveal the effects of ART per se. Through this approach, several evidences of long-lasting ART consequences have been reported in properly designed animal models [3,7,8,9,14,15,16], although the vast majority of these results has been evidenced in the mouse model. The renaissance of the laboratory rabbit as a reproductive model for human health is closely related to the growing evidence of periconceptional metabolic programming and its determining effects on offspring and adult health [17]. Moreover, many molecular/physiological events during human embryo development are more similar to those in rabbits than in mice, placing the rabbit as a pertinent animal model choice to study the impact of a disturbed periconceptional environment during ART and obtain results that are feasibly transferable to the clinic.

Until now, the vast majority of research has focused on improving birth rates after ART, and only a few groups are trying to discern whether these techniques leave a subtle legacy in offspring [18]. According to the last report from the European Society of Human Reproduction and Embryology (ESHRE), one of the steepest increases in treatment numbers was observed in the transfer of cryopreserved embryo (+13.6%), placing this technique as the second most commonly used in fertility treatments [19]. This indispensable tool in infertility laboratories maximizes the efficacy of ovarian stimulation cycles by allowing storage of the excess embryos and their later use, as well as enabling fertility preservation [20]. However, while the vast majority of studies mentioned above evaluated the effects of techniques that try to imitate physiological conditions, such as in vitro fertilization (IVF) or in vitro culture (IVC), cryopreservation procedures involve gamete/embryo exposure to potentially toxic cryoprotectant solutions and sub-zero temperatures [20]. Our recent study has demonstrated that each technique involved in a vitrified embryo transfer procedure (i.e., embryo vitrification-warming and embryo transfer) has an additive effect over the short- and long-term offspring development [15]. Then, negative synergies can exist when more than one stressor is present, with more severe preimplantation stress precipitating more deviant phenotypes [7,8,9,15]. Thereby, in most studies, all negative effects are considered jointly as part of the ART protocol. In this context, arguably the most pressing question in the DOHaD field is identifying how the molecular changes occurring secondary to ART exposures alter the growth and metabolic trajectories across the life course, to better know their biological relevance. Based on several omics approaches, ART practitioners are increasingly trying to elucidate the molecular mechanisms whereby these developmental changes arise after embryo manipulation (reviewed in [7,21]). Therefore, here we develop a rabbit model to assess the synergic effects of whole in vitro manipulations during a vitrified embryo transfer procedure, as clinical operation, studying both its phenotypic and molecular long-term effects.

## 2. Materials and Methods

Californian-bred rabbits were used throughout the experiment [22]. The experiment was approved by the “Universitat Politècnica de València” Ethical Committee prior to initiation of the study (research code: 2018/VSC/PEA/0116). The experiment was performed in accordance with relevant guidelines and regulations (Directive 2010/63/EU EEC). Animal experiments were conducted in an accredited animal care facility (code: ES462500001091).

### 2.1. Vitrified Embryo Transfer Procedure

Embryo vitrification and warming steps were adapted from the highly efficient protocol developed previously to cryopreserve rabbit embryos by vitrification [23,24]. This protocol allows the survival of >80% of the thawed embryos, having generated thousands of descendants in our laboratory since its implementation [24]. Briefly, vitrification was achieved in two steps. In the first step, embryos were placed for 2 min in a solution consisting of 10% (v/v) dimethyl sulfoxide (DMSO) and 10% (v/v) ethylene glycol (EG). In the second step, embryos were suspended for 1 min in a solution of 20% DMSO and 20% EG. Then, embryos suspended in vitrification medium were loaded into 0.125 mL plastic straws (French sterile ministraw; IMV Technologies, L’Aigle, France), which were sealed and plunged directly into liquid nitrogen to achieve vitrification. For warming, embryos were placed in 2 mL of 0.33 M sucrose at 25 °C to remove cryoprotectants and washed 5 min later. After warming, embryos were bilaterally transferred into the oviducts of pseudopregnant foster mothers by laparoscopy (10–14 embryos per female), following the protocol described by Besenfelder and Brem [25]. Ovulation was induced with an intramuscular dose of 1 µg of buserelin acetate (Suprefact, Hoechst Marion Roussel S.A, Madrid, Spain) 68–72 h before transfer. Briefly, foster mothers were anaesthetized with xylazine (5 mg/kg; Rompun; Bayern AG, Leverkusen, Germany) intramuscularly and ketamine hydrochloride (35 mg/kg; Imalgene 1000; Merial S.A, Lyon, France) intravenously and placed in Trendelenburg’s position. Then, embryos were loaded in a 17G epidural catheter, which was inserted through a 17G epidural needle into the inguinal region. Finally, monitoring the process by single-port laparoscopy, the catheter was introduced in the oviduct through the infundibulum to release the embryos. Both embryo vitrification and transfer processes used in this experiment were described in detail recently [24].

### 2.2. Experimental Design

Figure 1 illustrates the experimental design. Two experimental groups were developed and compared to study the effects of the entire vitrified embryo transfer (VET) operation: one from vitrified embryos transferred to the foster mothers (vitrified-transferred group: VT) and the other without any embryo manipulations (naturally conceived group: NC). Non-consanguineous healthy adult males and females with proven fertility were used to constitute both experimental groups. A total of 27 females were inseminated with the semen of unrelated males. Ovulation was induced with an intramuscular injection of 1 µg of buserelin acetate (Suprefact, Hoechst Marion Roussel S.A, Madrid, Spain). After three days, embryos from 13 females were recovered, vitrified-warmed and transferred into 13 foster mothers. Finally, a total of 158 vitrified-warmed embryos were transferred (95.3% survival rate after warming), generating 69 VT animals at birth (average litter size of 5.3 ± 0.64). Meanwhile, the NC group was generated using the remaining 14 females, which were maintained without any embryonic manipulation until the day of parturition. A total of 77 NC animals were obtained at birth (average litter size of 5.5 ± 0.62). The average litter sizes are in line with previous studies using this rabbit line [26]. On the day of birth, offspring were weighed, sexed and microchipped. Until nine weeks of age, animals were randomly distributed and caged collectively (eight rabbits per cage), and then were individually kept in separate cages (flat deck indoor cages: 75 × 50 × 40 cm). Until adulthood (20 weeks of age), both sexes were followed to determine the effects of the VET over the postnatal growth. In late adulthood (56 weeks of age), several organs were obtained and weighed from male animals (organ weight study). Females were excluded from this study to reduce confounding factors, as females are intrinsically more variable than males due to their cyclical reproductive hormones, which may interfere in the body condition [27]. Liver tissue samples were also collected due to the liver’s crucial role in growth and organ development from the fetal stage [28,29]. Then, a more in-depth study was done on this tissue, studying its proteomic signature to find molecular cues related to the phenotypic variations after VET.

### 2.3. Growth Performance during Postnatal Development

A total of 65 males (30 from VT and 35 from NC group) and 46 females (21 from VT and 25 from NC group) were weighed weekly from 1 to 20 weeks of age. In each week, the body weight differences between the experimental groups were evaluated.

### 2.4. Adult Body Weight and Organ Phenotypic Comparison

Males were euthanized by barbiturate overdose (125 mg/kg) injected intravenously at week 56 (late adulthood), when the growth plate was closed. Then, the body weight, organ (liver, lungs, heart, kidneys, adrenal glands, spleen and gonads) weights and fat tissue (perirenal and scapular) weight were determined. 

### 2.5. Determination of Hematological and Biochemical Parameters of Peripheral Blood in Adulthood

Before euthanasia, 40 individual blood samples (20 from VT and 20 from NC animals) were taken from the central ear artery. Animals were selected randomly, keeping 1–2 animals of each litter (parity) within each experimental group. From each animal, two blood samples were taken. The first was dispensed into an EDTA-coated tube (Deltalab S.L., Barcelona, Spain) and the other into a serum-separator tube (Deltalab S.L., Barcelona, Spain). Blood count was performed from EDTA tubes at most 10 min after the collection using an automated veterinary hematology analyzer MS 4e automated cell counter (MeletSchloesing Laboratories, Osny, France) according to the manufacturer’s instructions. The blood parameters recorded were as follows: white blood cells, lymphocytes, monocytes, granulocytes, red blood cells, hemoglobin and hematocrit. From the second tube, biochemical analyses of the serum glucose, cholesterol, albumin, total bilirubin and bile acids were performed as hepatic metabolic indicators. Briefly, samples were immediately centrifuged at 3000× *g* for 10 min, and serum was stored at −20 °C until analysis. Then, glucose, cholesterol, albumin and total bilirubin levels were analyzed by enzymatic colorimetric methods, while bile acids were measured by photometry. All the methodologies were performed in an automatic chemistry analyzer model Spin 200E (Spinreact, Girona, Spain), following the manufacturer’s instructions. All samples were processed in duplicate.

### 2.6. Statistical Analysis of Phenotypic Data

A general linear model (GLM) was fitted for the analysis of body weight in each week, organ weights and peripheral blood parameters, including the experimental group as fixed effect, and the biological and foster mother as random effects. Litter size was used as covariate for body weight correction since birth until adulthood, although it remained non-significant from the third week of age. In the case of organ weights, data were corrected using body weight as a covariate. The growth rate was estimated by nonlinear regression using the Gompertz curve equation, well suited for rabbits [15,30]: y = a exp(−b exp(−kt)), where y is the observed body weight of one individual at a specific age (t). The rest of the parameters (a, b and k) of the Gompertz function have a biological interpretation, k being the parameter related to the rate of maturing (growth rate). The growth curves were plotted using the estimated parameters. Data were expressed as means ± standard error of means. Differences of *p* < 0.05 were considered significant. All statistical analyses were performed with SPSS 21.0 software package (SPSS Inc., Chicago, IL, USA).

### 2.7. Comparative Proteomic Analysis: Sampling, Protein Extraction and Quantification

A total of 8 individual samples (4 VT and 4 NC) were taken from adult rabbit males (selected randomly from different litters), retrieving some liver biopsies from the same organ (one individual, one sample). The uniformity of the liver tissue (four major cell types, of which hepatocytes constitute ≈70% of the total liver cell population) facilitates the sampling and data interpretation [31]. The samples were immediately washed with a phosphate-buffered saline solution to remove blood remnants and were directly flash-frozen in liquid nitrogen and stored at −80 °C for the proteomic study. The proteome analyses were performed in the Proteomics Unit of the University of Valencia, Valencia, Spain (member of the PRB2-ISCIII ProteoRed Proteomics Platform). Proteins from liver biopsy were extracted in Lysis buffer (7M Urea, 2M thiourea, 4% CHAPS, 30 mM Tris pH 8.5) using 2D Grinding Kit (GE Healthcare, Chicago, IL, USA). Samples were quantified by an RC_DC kit (BioRad, Hercules, CA, USA) according to the manufacturer’s instructions. 

### 2.8. Complete Proteome: Spectral Library Building by In-Gel Digestion and LC-MS/MS—Data-Dependent Acquisition Analysis

First, we conducted a data-dependent acquisition (DDA) analysis to study the complete proteome by building up a spectral library using in-gel digestion and LC-MS/MS. The complete proteome was analyzed from a pool obtained by mixing aliquots with an equivalent amount of all the samples (100 µg in total) to build the spectral library from a 1D SDS PAGE gel. The career of the gel corresponding to the library was cut into three pieces, which were then digested with sequencing grade trypsin (Promega), as described elsewhere [32]. Gel slides were digested at 37 °C using 500 ng of trypsin. The digestion was stopped with 10% trifluoroacetic acid (TFA), and the supernatant (SN) was removed. Then, the library gel slides were dehydrated with pure acetonitrile (ACN). The new peptide solutions were combined with the corresponding SN. The peptide mixtures were dried in a speed vacuum and resuspended in 100 μL 2% ACN (v/v); 0.1% TFA (v/v). Five microliters of the digested fragments was loaded into a trap column (3 μm particle size C18-CL, 350 µm diameter × 0.5 mm long; Eksigent Technologies) and desalted with 0.1% TFA (v/v) at 3 µL/min for 5 min. The peptides were loaded into an analytical column (LC Column, 3 µm C18-CL, 75 um × 12 cm, Nikkyo), and equilibrated in 5% ACN 0.1% FA (formic acid) (v/v). Peptide elution was carried out with a linear gradient of 5 to 35% B for 90 min (A: 0.1% FA (v/v); B: ACN, 0.1% FA (v/v)) at a flow rate of 300 nL/min. Peptides were analyzed in a mass spectrometer nanoESI qQTOF (5600 TripleTOF, ABSCIEX, Alcobendas, Madrid, Spain). The tripleTOF was operated in information-dependent acquisition mode, in which a 250 ms TOF MS scan from 350 to 1250 m/z was performed, followed by 150 ms product ion scans from 350 to 1500 m/z on the 25 most intense 2–5 charged ions. The rolling collision energy equations were set for all ions as for 2+ ions according to the following equations: |CE| = (slope) × (m/z) + (intercept).

### 2.9. LC-SWATH-MS Acquisition: Analysis of Individual Samples

After that, analysis of individual samples was carried out by LC-SWATH-MS acquisition. To determine quantitative differences in liver protein composition among our experimental rabbit progenies, the SWATH analysis of individual liver samples was performed. Thus, 20 µg of total protein extracted from each sample was loaded in the 1D SDS PAGE, and the carrier corresponding to each sample was digested with sequencing-grade trypsin (Promega), as described elsewhere [32], using 500 ng of trypsin for each sample, and digestion was set to 37 ºC. The trypsin digestion was stopped with 10% TFA (v/v), the SN was removed and the library gel slides were dehydrated with pure ACN. The new peptide solutions were combined with the corresponding SN. The peptide mixtures were dried in a speed vacuum and resuspended in 25 µL of 2% ACN (v/v); 0.1% TFA (v/v). Five microliters of each sample was loaded into a trap column (3 μm particles size 18-CL, 350 µm diameter × 0.5 mm long; Eksigent Technologies) and desalted with 0.1% TFA (v/v) at 3 μL/min for 5 min. The peptides were loaded into an analytical column (LC Column, 3 μm C18-CL, 75 μm × 12 cm, Nikkyo), equilibrated in 5% ACN (v/v) 0.1% FA (v/v). Peptide elution was carried out with a linear gradient of 5 to 35% B in 120 min (A: 0.1% FA (v/v); B: ACN, 0.1% FA (v/v)) at a flow rate of 300 nL/min. The tripleTOF was operated in SWATH mode, in which a 0.050 s TOF MS scan from 350 to 1250 m/z was performed, followed by 0.080 s product ion scans from 350 to 1250 m/z (3.05 s/cycle). The SWATH windows used were as follows: 15 Da window widths from 450 to 1000 Da, 37 windows. Collision energy was set to optimum energy for a 2+ ion, and the mass spectrometer was always operated in high-sensitivity mode.

### 2.10. Protein Identification, Validation and Quantification

After library LC-MS/MS, the SCIEX.wiff data-files were processed using ProteinPilot v5.0 search engine (AB SCIEX, Alcobendas, Madrid, Spain). The Paragon algorithm (5.0.2.0, 5174) of ProteinPilot was used to search against the Uniprot Mammalia protein sequence database (1,376,814 proteins searched) with the following parameters: trypsin specificity, cys-alkylation, without taxonomy restriction, and the search effort set to through and false discovery rate (FDR) correction for proteins [33]. To avoid using the same spectral evidence in more than one protein, the identified proteins were grouped based on MS/MS spectra by the Protein-Pilot Pro GroupTM Algorithm, regardless of the peptide sequence assigned. The protein within each group that could explain the most spectral data with confidence was depicted as the primary protein of the group. The resulting Protein-Pilot group file was loaded into PeakView^®^ (v2.1, AB SCIEX, Alcobendas, Madrid, Spain), and peaks from SWATH runs were extracted with a peptide confidence threshold of 95% confidence and a FDR less than 1%. The number of peptides per protein was set at 50, and six transitions per peptide were necessary to quantify one peptide. Modified peptides were excluded. After peptide detection, peptides were aligned among different samples using peptides detected at high confidence from the library. The extracted ion chromatograms were integrated, and the areas were used to calculate the total protein quantity. The mass spectrometry proteomics data were deposited with the ProteomeXchange Consortium via the PRIDE [34] partner repository with the dataset identifiers PXD016677 (SWATH data) and PXD016874 (Spectral Library data).

### 2.11. Statistical Analysis of the Proteome and Functional Annotation of the Differentially Expressed Proteins

The quantitative data obtained by PeakView^®^ were analyzed using MarkerView^®^ (v1.2, AB SCIEX, Alcobendas, Madrid, Spain). Normalization of the calculated areas was done by summing total areas. A t-test was used to identify the differentially expressed proteins (DEPs) among the two experimental groups (VT and NC). Proteins were considered differentially expressed with an adjusted *p*-value ≤0.05. Rabbit (*Oryctolagus cuniculus*) identifiers were obtained using the Blast tool from UniProt, keeping the output with the high-identity score. Principal component analysis (PCA) and Heat-Map (HM) clustering were performed using ClustVis (https://biit.cs.ut.ee/clustvis/). Functional descriptive pie charts of DEPs were obtained from the Panther web tool (http://pantherdb.org/) using *Homo sapiens* as a reference and the human orthologous gene names (obtained from Biomart-Ensembl web tool: https://www.ensembl.org/info/data/biomart/index.html) as input data. Functional annotation of DEPs, enrichment analysis of their associated “Gene Ontology” (GO) terms and “Kyoto Encyclopedia of Genes and Genomes” (KEGG) pathways analysis were computed using the free available bioinformatics software DAVID Functional Annotation Tool (https://david.ncifcrf.gov/home.jsp; version 6.8), considering a *p*-value (modified Fisher’s exact test, EASE score) of less than 0.05.

## 3. Results

### 3.1. Animals Derived after Vitrified-Thawed Embryo Transfer Procedure Exhibit Higher Birth Weight, but Lower Growth Performance, until Adulthood

At parturition, animals from the vitrified-transferred (VT) group showed higher birth weight than those from the naturally conceived (NC) group. There was no interaction between treatment and sex. Mean birth weights were 67.8 ± 1.46 and 60.5 ± 1.72 g for VT and NC males, respectively (*p* < 0.05), using the covariate litter size (6.9 ± 0.38, significant effect at *p* < 0.05). In the case of females, the mean birth weights were 63.3 ± 1.71 and 58.1 ± 1.45 g for VT and NC groups, respectively (*p* < 0.05), using the covariate litter size (6.7 ± 0.28, significant effect at *p* < 0.05).

However, VT animals showed reduced growth performance until adulthood compared to those NC (Figure 2). Hence, the parameters governing the Gompertz growth curve established that the growth velocity (k parameter) was lower in the VT compared with the NC group, both in males (0.16 ± 0.005 VT vs. 0.20 ± 0.007 NC, *p* < 0.05, Figure 2A) and females (0.17 ± 0.005 VT vs. 0.21 ± 0.004 NC, *p* < 0.05, Figure 2B). Therefore, weaned animals showed significant mean weight differences between groups (NC-VT ± standard error) from 5 to 9 weeks of age: 248.0 ± 20.98 g for males (Figure 2C) and 246.8 ± 56.11 g for females (Figure 2D). From this age, the mean weight differences between groups still increased until adulthood (from 10 to 20 weeks of age), being 724.6 ± 117.54 g for males (Figure 2C) and 466.4 ± 113.64 g for females (Figure 2D).

### 3.2. At Adulthood, Animals Derived from Vitrified-Transferred Embryos Showed Lower Body Weight and Reduced Weight in Some Vital Organs

At 56 weeks of age, VT males showed lower (7.0%) body weight compared to NC counterparts (Table 1). No differences were observed in the body composition in terms of fat tissue amount (Table 1). Moreover, the VT group showed lower liver (9.1%) and heart (13%) weights, even after data were corrected by body weight (Table 1). No significant weight differences were observed for the rest of the analyzed organs.

### 3.3. The Peripheral Blood Parameters (Healthy Status) of the Vitrified-Transferred Progeny Were Similar to Those of the NC Group

As shown in Table 2, there were no significant differences in the peripheral blood cells profile (white blood cells, red blood cells, hemoglobin and hematocrit) between NC and VT rabbit males. Attending to the serum biochemical data, higher levels of albumin and lower levels of cholesterol were exhibited by the VT animals (*p* < 0.05). However, these levels ranged between the normal values in rabbits [35,36].

### 3.4. The Liver Protein Profile Was Influenced by Vitrified-Thawed Embryo Transfer Procedure

The complete spectral library included 28,685 MS/MS spectra corresponding to 14,737 distinct peptides and 1846 proteins with a FDR ≤1%. With the restrictions used for extraction parameters of the areas, 1491 proteins (FDR <1%) were quantified in the eight samples. Protein data analysis identified 77 differentially expressed proteins (DEPs) in *Oryctolagus cuniculus* taxonomy. PCA and Heat-Map analysis showed that, despite expected individual variability, samples from each group were clustered together (Figure 3). From these DEPs, there was a higher number of downregulated (56/77, 72.7%) than upregulated (21/77, 27.3%) in VT samples compared to the NC group. Of the proteins that were significantly different, a total of 66 DEPs were recognised by the DAVID software. Notably, 80% of these DEPs had catalytic/binding activity and 60% were involved in cellular/metabolic process, mainly (76%) in cell and organelles (Appendix A). Annotation of DEPs and the fold change values obtained are shown in Appendix A. Functional GO term enrichment and KEGG pathway analysis of DEPs were recorded in Appendix A. The most relevant metabolic alteration denoted by the protein profile was that related to oxidative phosphorylation, suggesting an impaired oxidative metabolism in the mitochondria. Furthermore, hints of dysregulation in zinc (Zn) and lipid metabolism were identified.

## 4. Discussion

Here, we describe for the first time an integrative study that evaluates and correlates the long-term consequences of the entire vitrified embryo transfer (VET) procedure at the phenotypic and proteomic levels. We observed that individuals born after VET showed a diminished growth rate and lower vital organ weights. Moreover, our findings derived from the hepatic proteomic analysis revealed a significant metabolic reprogramming. Altogether, our results represent firm evidence of the developmental plasticity exhibited by mammalian embryos in response to in vitro stressors, which results in irreversible phenotypic and developmental variations. 

In humans, the first baby born after VET was born only 36 years ago, so the long-term impact of this procedure remains unknown [37]. Here, long-term effects after VET have been demonstrated even though our model is a simplified reality of the usual human process. Today, VET procedures are a primary component of ART cycles [19], and although it can be lethal to some embryos, this technique was considered neutral for survivors for a long time [38]. However, decades ago, some studies began to doubt the innocuousness of this technique [16,38,39]. Recently, we have proven that embryo cryopreservation has an additive effect over those attributable to the ex vivo embryo handle during the transfer procedure, both in the short- and long-term offspring development [15]. As in vitro embryo handling and transfer are mandatory techniques of the VET clinical service [24], here we compared VT vs. NC animals to comprise the consequences of the whole VET protocol. Supporting this proposal, it has been reported that when different stressors are present, these can act synergistically inducing more adverse effects [7,8,9,15]. Thereby, in most studies, all adverse effects are commonly considered jointly as part of the ART protocol [9]. First of all, we observed that VT animals exhibited higher body weight at parturition, which was consistent with our previous findings [15,40] and the health outcomes reported in humans [41,42] after VET. Intriguingly, the most dramatic ART effect in cattle and sheep is known as large offspring syndrome, characterised by a large size at birth and increased birth weight [9,43]. Imprinting modifications of some growth-related genes have also been suggested to explain these observed variations [9,30,41,43]. 

However, although birth deviations are usually re-established through compensatory growth in rabbits [44], VT animals showed a diminished body weight throughout life in comparison with NC animals. Furthermore, we observed that these weight differences were maintained until adulthood, although the body fat was similar between the two experimental groups. Therefore, differences in body weight confirmed a lower growth performance in VT animals, which agreed with previous studies carried out in rabbits undergoing VET [15,30,39]. Concordantly, previous studies have demonstrated that the nature of the preimplantation environment during ART can program development, affecting postnatal growth and phenotype [7,14,45,46,47]. Notably, although the pattern of postnatal growth alteration in VT animals was similar in both sexes, males duplicated body weight differences between VT and NC compared to the females at the age of 20 weeks. During preimplantation development, male and female embryos may display phenotypic differences that can be attributed to their different sex chromosome complements [48]. This sex asymmetry reflects that preimplantation embryos are already poised to respond differentially to environmental changes in a sex-specific fashion [46,48]. It may explain the frequent sex bias observed in various models of DOHaD and developmental reprogramming after ART [7,16,46,49]. Based on this evidence, we hold that the postnatal growth trajectory after VET in Californian rabbits is sexually dimorphic, which is in contrast with our previous data derived from New Zealand rabbits [15]. Supporting this, it has been reported that several phenotypes linked to ART are condition-specific and sexually dimorphic, but also strain-specific [7]. Phenotypic variation by strain could be attributed to differences in both genetic code and regulatory mechanisms (e.g., epigenetics), which depends on genetic background. Therefore, the strain is a critical variable in how embryos might differentially respond to similar ART conditions and display different postnatal phenotypes. In this article, we also have shown that animals born after VET exhibit lower liver and heart weights. Alterations in the same organs have been previously described after IVC and VET [30,50,51]. As ART, including embryo cryopreservation [40,52], has been associated with placental alterations, it could be the reason explaining liver disturbances after VET, as decreased materno-fetal nutrition during gestation induces reduced liver mass and perturbed liver function [53]. Besides, liver disturbances could affect the heart, given the strong interaction between both organs’ physiologies [54].

The role of the liver in organ growth and development is well documented [28,29]. As the ‘omics’ sciences are used to describe the flow of biological information in an organism, we examined the hepatic proteomic signature to find out an explanation for the phenotypic differences after VET. Functional analysis revealed five DEPs (*NDUFB9, NDUFB8, NDUFA10, ATP5A1, ATP6V1A*) involved in oxidative phosphorylation (OXPHO) with an average abundance two times lower in VT animals, suggesting a lower OXPHO activity. As mitochondrial OXPHO is the primary source of ATP to support life and development, OXPHO troubles are found in most diseases related to growth alteration, short stature and dwarfism [55,56,57,58,59], which could explain the diminished growth in VT animals. Concordantly, Feuer et al. [14] also demonstrated that IVF and IVC promoted postnatal growth reduction, correlated with mitochondrial dysfunction and systemic oxidative stress. Evidence of mitochondrial and OXPHO dysfunction has been described in embryos after IVF and cryopreservation [8,60], as well as in adult IVF livers [7]. Here, we provide the first proteomic evidence of OXPHO alterations in adult livers after VET. Besides, we found upregulation in four DEPs (*GSTM2, LOC100357148, ADH2-1, EPHX1*) related to the detoxification mechanisms of cytochrome P450, with the first two having glutathione transferase activity. Glutathione transferase activity is one of the main cellular mechanisms to tackle the damage from reactive oxygen species and, together with the P450 enzymes, are involved in the hepatic drug metabolism pathways [61,62]. Then, the upregulation of these proteins could be indicative of high hepatic detoxification requirements in an organism with oxidative metabolic disturbances. We can, therefore, envision a cardinal role for mitochondria in the reprogramming mechanism by which preimplantation stressors induce postnatal alterations, since oxidative stress seems to be ubiquitously present in ART tissues [7].

An in-depth analysis revealed five (*CNBP, LASP1, DPP3, LOC100339065, PMPCB*) downregulated Zn-binding proteins in VT animals. It is known that transcriptional mechanisms are present to reduce gene expression of Zn-binging proteins when zinc is limiting, thus conserving Zn for more essential functions [63]. Therefore, these data suggested a weakened Zn metabolism in VT animals, which could be related with its diminished growth performance. Supporting this, a recent study has reported that serum Zn content was lower in ART children, who were shorter than the naturally conceived counterparts, as Zn is required for healthy growth [64]. In addition, we recorded seven DEPs (*APOB, ATP5A1, STARD10, FABP1, ADH2-1, AMBP, RBP4*) involved in lipid and fatty acid metabolism. There is evidence demonstrating ART-induction of modifications in the lipid metabolism in embryonic [8], fetal [65] and adult [14,49] stages, suggesting that ART offspring were less efficient in their use of internal lipids for ATP production, probably due to altered mitochondrial function. Particularly, associated functional terms revealed that *APOB* participates in the regulation of the cholesterol biosynthetic process and its metabolism, so its downregulation after VET might explain the lower serum cholesterol levels exhibited by VT animals. This fact agreed with previous studies reporting disturbances of steroid metabolism in the placenta, fetal liver and adult serum and tissues (including liver) after ART [8,14,66,67].

However, despite some phenotypic and physiological changes appearing in VT animals, this progeny appeared healthy, sustained by the peripheral blood results in adulthood. Although some serum markers of hepatic function varied significantly between VT and NC, it remained within the standard values in rabbits. Besides, no striking difference was detected either during the management of both VT and NC animals or during the dissection study. Therefore, as no signs of disease were detected in VT animals, we hold that differences observed in the VT offspring can be partially attributed to a possible selection of cryo-resistant embryos, which originate a subpopulation with intrinsic differences compared to the general population [15]. In addition, it is well accepted that ART incurs environmental stressors during embryo development, triggering embryonic response mechanisms that can result in phenotypic variation later in life. These variations are attributed to the embryonic developmental plasticity [3,4], which refers to the capacity of a genotype to produce different phenotypes in response to environmental changes, contributing to diversity among individuals, populations and species. Reprogramming and developmental plasticity are believed to be mediated by epigenetic changes, which can persist into subsequent generations [6,13]. Therefore, future studies should determine the transgenerational inheritance of the VET effects.

## 5. Conclusions

In conclusion, in vitro embryo manipulation throughout vitrification and transfer procedures causes long-term effects on growth rate, body weight and vital organ weights at adulthood. Furthermore, molecular data obtained from the hepatic tissue are related to these phenotypic changes, paving the way to finding molecular markers and pathways that enhance the knowledge of long-lasting ART effects, their detection and their health relevance. With more novel ARTs waiting on the horizon, this study should represent a significant step towards promoting a paradigm shift in the characterization of long-term consequences of ARTs in adulthood. Taking into account that the long-term effects of ART are specific to each procedure, tissue, sex or species, a systems biology perspective might be the way to elucidate the adaptive mechanisms of embryos during ART procedures.

## Figures and Tables

**Figure 1 animals-10-01043-f001:**
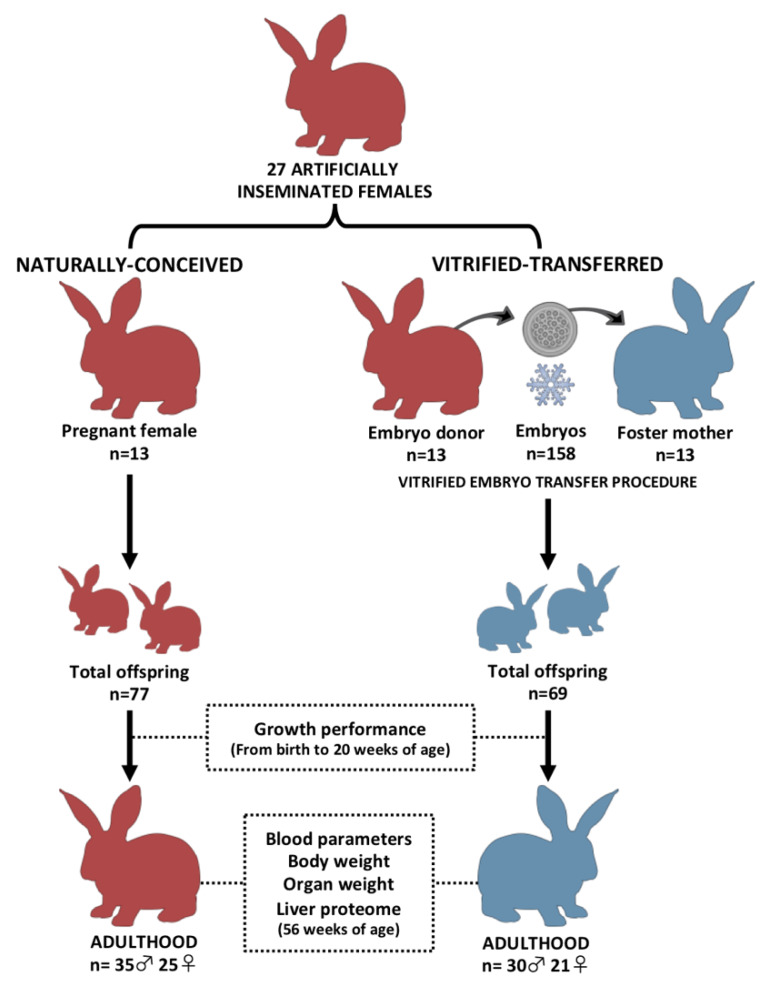
Experimental design. Two experimental progenies were developed and compared during development and in adulthood. Vitrified-transferred population arises from vitrified embryos transferred into foster mothers; meanwhile, the naturally conceived population was generated without embryo manipulations. In both groups, after parturition day, offspring were weighed every week until adulthood (growth performance). In adulthood, peripheral blood parameters and organ weight comparisons were performed in male animals. From liver samples, a proteomic comparative analysis was developed. n: number of animals.

**Figure 2 animals-10-01043-f002:**
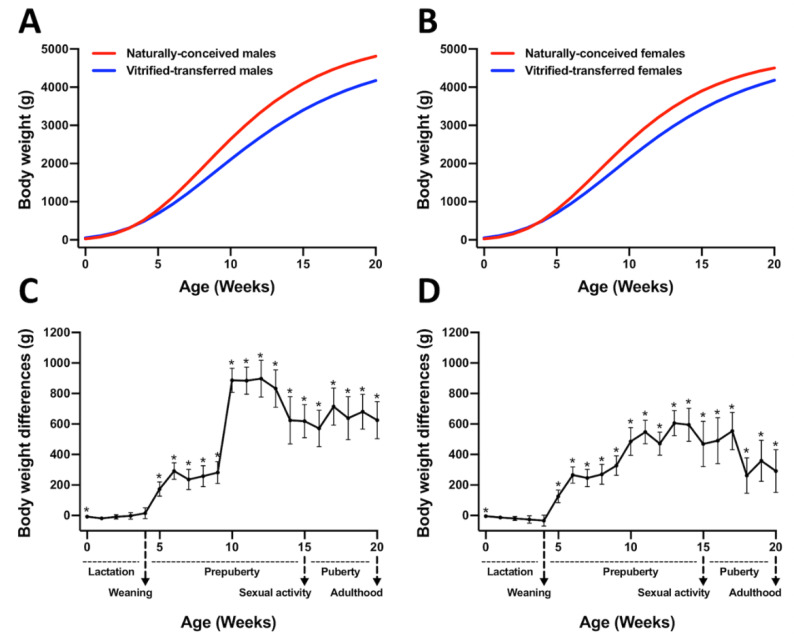
Growth curves and differences in body weight between animals derived after vitrified embryo transfer procedure (VT) and those naturally conceived (NC). Growth curves were fitted using the Gompertz equation for (**A**) males and (**B**) females, comparing NC and VT groups. Differences in body weight for (**C**) males and (**D**) females were computed as NC-VT. Asterisks denote significant differences.

**Figure 3 animals-10-01043-f003:**
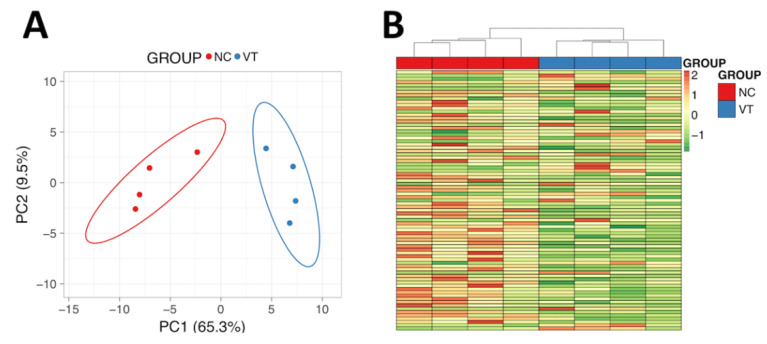
Molecular analysis of male liver samples obtained from adult males derived from vitrified-transferred embryos (VT) and naturally conceived animals (NC). (**A**) Principal Component Analysis. (**B**) Heat-Map clustering. The representation of sample variability between the experimental groups was performed, taking into account only the differentially expressed proteins. PC1: Principal Component 1. PC2: Principal Component 2.

**Table 1 animals-10-01043-t001:** Body weight and dissection data of adult, naturally conceived males and those derived from vitrified-transferred embryos.

Traits	Naturally Conceived (n = 35)	Vitrified-Transferred (n = 30)
Body weight (Kg)	5.7 ± 0.10 ^a^	5.3 ± 0.11 ^b^
Perirenal fat (g)	173.7 ± 11.09	184.2 ± 12.13
Scapular fat (g)	58.9 ± 5.32	66.1 ± 5.99
Kidneys (g)	22.7 ± 0.45	22.3 ± 0.42
Liver (g)	102.1 ± 2.51 ^a^	92.8 ± 2.37 ^b^
Spleen (g)	1.4 ± 0.08	1.3 ± 0.07
Lungs (g)	25.6 ± 1.20	26.7 ± 1.13
Heart (g)	13.1 ± 0.40 ^a^	11.4 ± 0.42 ^b^
Gonads (g)	6.2 ± 0.36	6.9 ± 0.34
Adrenal glands (g)	0.6 ± 0.04	0.7 ± 0.03

n represents the number of animals. All organ/tissue weights were corrected using the body weight as a covariate. Data are expressed as least-squares mean ± standard error of means. a, b Values in the same row with different superscripts are significantly different (*p* < 0.05).

**Table 2 animals-10-01043-t002:** Hematological and biochemical comparison between peripheral blood of adult, naturally conceived males and those derived from vitrified-transferred embryos.

Peripheral Blood Parameters	Naturally Conceived (n = 20)	Vitrified-Transferred (n = 20)
**Hematology**		
White blood cells (10^3^/mm^3^)	9.7 ± 0.77	8.7 ± 0.77
Lymphocytes (%)	42.6 ± 3.14	41.8 ± 3.14
Monocytes (%)	4.2 ± 0.844	5.1 ± 0.844
Granulocytes (%)	42.5 ± 2.71	45.9 ± 2.71
Red blood cells (10^6^/mm^3^)	6.0 ± 0.14	6.1 ± 0.14
Hemoglobin (g/dL)	12.5 ± 0.32	12.7 ± 0.32
Hematocrit (%)	42.4 ± 1.27	42.3 ± 1.27
**Serum metabolites ^+^**		
Albumin (g/dL)	4.2 ± 0.05 ^b^	4.4 ± 0.05 ^a^
Bile acids (µmol/L)	7.2 ± 0.87	6.9 ± 0.87
Cholesterol (mg/dL)	40.1 ± 2.09 ^a^	31.7 ± 2.09 ^b^
Glucose (mg/dL)	127.7 ± 9.41 ^b^	141.4 ± 9.41 ^a^
Bilirubin total (mg/dL)	0.1 ± 0.01	0.1 ± 0.01

n represents the number of animals; data are expressed as least-squares means ± standard error of means. a, b Values in the same row with different superscripts are significantly different (*p* < 0.05). ^+^ Serum indicators of the hepatic function.

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
