# Peer review of "Long-Term Phenotypic and Proteomic Changes Following Vitrified Embryo Transfer in the Rabbit Model"

_animals, 2020, doi:10.3390/ani10061043_

Round 1

Reviewer 1 Report

The aim of the study was to evaluate the long-term effects of embryo vitrification and transfer procedures (VET). Using the rabbit as an animal model, the authors analysed first the post-natal weight and growth performance. Second, in the adult offspring, the authors analysed the body and internal organs weights; haematological and biochemical parameters in peripheral blood; and the liver proteome. They found important differences in birth weight, growth performance, body, liver and heart weights. They also observed significant differences in specific liver proteins related to oxidative phosphorylation and zinc and lipid metabolism.

One of the strongest points of the manuscript is that the authors used a large number of vitrified transferred embryos (n=158), offspring (n=69) and males offspring for specific analyses (n=30), making their observations very robust. Moreover, the authors included a broad range of techniques to evaluate phenotypical and molecular changes in the adult offspring. The results of this study are very important for both animal breeding programs and human fertility clinic.

Minor comments:

Line 297: Remove word «After» at the beginning of the sentence.

Line 304: Remove “Figure 2A and 2B” and mention the figures right after its description (line 306).

Figure 3: It would be nice for the reader if authors include in “Figure 3” which differences are significant (P<0.05).

Line 337: I do not understand why the authors named the section: “Vitrified-thawed embryo transfer procedure seems to be neutral on the peripheral blood parameters”. What does exactly mean “seems to be neutral”? Can the title be rephrased to something more specific?

Line 338: Remove word “Even” at the beginning of the sentence.

Line 449: Authors should change the UNIPROT accession number by the official names of the proteins. Same for lines: 465, 474, 482, 489.

Author Response

REMARK 1. Line 297: Remove word «After» at the beginning of the sentence.

It has been done.

REMARK 2. Line 304: Remove “Figure 2A and 2B” and mention the figures right after its description (line 306).

This suggestion has been addressed.

REMARK 3. Figure 3: It would be nice for the reader if authors include in “Figure 3” which differences are significant (P<0.05).

According to reviewer 2, Figure 3 has been removed, as it only contains the “Box plots” showing the raw data distributions of the phenotypic traits before any statistical analysis. The idea was to show how the crude data are distributed, but we also considered that it is not essential information and could deviate the lecturers from the main results.

REMARK 4. Line 337: I do not understand why the authors named the section: “Vitrified-thawed embryo transfer procedure seems to be neutral on the peripheral blood parameters”. What does exactly mean “seems to be neutral”? Can the title be rephrased to something more specific?

We are totally in agreement with the reviewer, as this title could induce confusion. What we would say is that peripheral blood parameters between VT and NC progenies were similar. Therefore, we change the title by the following one: “The peripheral blood parameters (healthy status) of the vitrified-transferred progeny were similar to those of the NC group”.

REMARK 5. Line 338: Remove word “Even” at the beginning of the sentence.

This suggestion has been addressed.

REMARK 6. Line 449: Authors should change the UNIPROT accession number by the official names of the proteins. Same for lines: 465, 474, 482, 489.

The UNIPROT accession numbers have been replaced by the official gene/protein names.

Reviewer 2 Report

The authors investigated the effects of vitrified embryo transfer procedure on long term phenotype of rabbits. The authors revealed that vitrified embryo transfer procedure induced higher body weight at birth, lower growth, and reprograming of liver proteome. The present results would provide valuable information to readers of this journal, however, there are some points to elucidated and revised for publication.

General points

  • I feel the manuscript, especially introduction and discussion section, is too long as an original paper (even like a review article) and redundant, containing much information from previous report which is not directly related to the present results. Please make a discussion based on the present results referring to related previous reports, not consisted of speculation and discussion about the previous reports.
  • The authors concluded that VET suppress growing in rabbit on ground of its weight of body, liver and heart. However, these can be changed by physiological or pathological conditions. As the value of serum cholesterol was notably low in VT group compared to NC group, it is possible that VT rabbits were lean which induce low body weight. To prove growth alteration, information about other body parameters including body size and composition are demanded. At least, discussion about such possibility would be essential.
  • What is the difference between VET (vitrified embryo transfer) and VT (vitrified-transferred)? Even the authors confound them: e.g. vitrification-transfer procedure (VET) in line 377. Those 2 words should be unified. In the same manner, what is the difference between vitrified-thawed embryos (line 126) and vitrified-warmed embryos (line 463)
  • I wonder if it is necessary to present both of Figure 3 and Table 1, which show the same data. There are no comments on Figure 3 neither in results nor discussion. Is it necessary to present Figure 3? If so, please explain the findings form the figure. Additionally, location of data form each group is opposite in Figure 3 to Table 1(NC on left side and VT on right side in Figure 3). It would be better to unify the side and color showing date from each group in every figure and table to avoid confusion for readers.

Specific points

  • Line 121: Information about mother (NC), donor and recipient (VT) rabbit is missing. Are there any differences in age, reproductive history etc. between the groups?
  • Line 133, adulthood: Specify the end point (age) of the study.
  • Line 159, French straw: Specify the model and manufacturer.
  • Line 160: Precise procedure for embryo warming, euthanasia and anesthesia is demanded. Please make a brief (at least) statement about the procedure, since not all readers access every referred report.
  • Line 161, 10-14 embryos per female: Which were they transferred into bilateral or unilateral oviduct?
  • Line 180, blood sample: Which condition was it collected in fasted or fed?
  • Line 180, 209: How were the sampled rabbits selected from 111 rabbits?
  • Line 342: Indicate reference(s) about normal values in rabbit.
  • Line 432, our previous data: Indicate reference(s).

Author Response

REMARK 7. I feel the manuscript, especially introduction and discussion section, is too long as an original paper (even like a review article) and redundant, containing much information from previous report which is not directly related to the present results. Please make a discussion based on the present results referring to related previous reports, not consisted of speculation and discussion about the previous reports.

All the manuscript, especially introduction and discussion, has been reviewed carefully, removing the redundant information and keeping only the essential studies to avoid speculation. We feel that the manuscript seems more concise and more easily readable, having been removed 11 non-essential references and more than 1100 words.

REMARK 8.  The authors concluded that VET suppress growing in rabbit on ground of its weight of body, liver and heart. However, these can be changed by physiological or pathological conditions. As the value of serum cholesterol was notably low in VT group compared to NC group, it is possible that VT rabbits were lean which induce low body weight. To prove growth alteration, information about other body parameters including body size and composition are demanded. At least, discussion about such possibility would be essential.

We are entirely in agreement with this point of view offered by the reviewer. One can’t be sure if VET incurred in a disturbing growth if additional body parameters are not measured. However, we have data generated during the dissection study and related with the fat content of both NC and VT animals, as we also measured the total fat amount in the perirenal region, as well as in the scapular one. Since we want to show data related to essential organs, we considered in the first instance that this data was not relevant, since it is not vital tissue. However, in light of this issue, we implemented and discussed these data in the manuscript, proving that similar content of fat (corrected by body weight) was exhibited by NC and VT animals, which confirms that weight differences were the reflection of growth differences, rather than differences in body composition.

REMARK 9. What is the difference between VET (vitrified embryo transfer) and VT (vitrified-transferred)? Even the authors confound them: e.g. vitrification-transfer procedure (VET) in line 377. Those 2 words should be unified. In the same manner, what is the difference between vitrified-thawed embryos (line 126) and vitrified-warmed embryos (line 463)

Effectively, there is a mismatch in the terminology and, although the idea was the same, the use of different words could induce confusions.  Really, VET is referred to the vitrified embryo transfer procedure (i.e. the whole assisted reproductive procedure), and vitrified-transferred (VT) is the abbreviation used for the embryos subjected to the VET process or the offspring derived from. Besides, as ice is not formed during vitrification, the correct word is warming, rather than thawing, and it has been corrected in the manuscript.

REMARK 10.  I wonder if it is necessary to present both of Figure 3 and Table 1, which show the same data. There are no comments on Figure 3 neither in results nor discussion. Is it necessary to present Figure 3? If so, please explain the findings form the figure. Additionally, location of data form each group is opposite in Figure 3 to Table 1(NC on left side and VT on right side in Figure 3). It would be better to unify the side and color showing date from each group in every figure and table to avoid confusion for readers.

In agreement with these comments and as commented above, Figure 3 has been removed, as it contains not essential information and could deviate the lecturers from the main results. However, colours and locations of the corresponding experimental groups have been unified throughout the manuscript to avoid confusions.

REMARK 11. Line 121: Information about mother (NC), donor and recipient (VT) rabbit is missing. Are there any differences in age, reproductive history etc. between the groups?

Healthy adult males and females with proven fertility were used to constitute both experimental groups. It has been included in the manuscript to avoid confusions.

REMARK 1. Line 133, adulthood: Specify the end point (age) of the study.

Rabbit animals reach adulthood at 20 weeks of age, then the growth performance study was addressed between birth and the 20th week. However, the dissection study for body and organ weighs was performed in late adulthood (56th week), to ensure that the growth plate is closed and the body conditions were stabilized. These endpoints were clarified in the manuscript.

REMARK 2. Line 159, French straw: Specify the model and manufacturer.

It has been implemented in the manuscript.

REMARK 3. Line 160: Precise procedure for embryo warming, euthanasia and anesthesia is demanded. Please make a brief (at least) statement about the procedure, since not all readers access every referred report.

 For warming, embryos were placed in 2 mL of 0.33 M sucrose at 25 °C to remove cryoprotectants and washed 5 min later.

 Males were euthanised by barbiturate overdose (125 mg/kg) injected intravenously.

 Foster mothers were anaesthetized with xylazine (5mg/kg; Rompun; Bayern AG, Leverkusen, Germany) intramuscularly and ketamine hydrochloride (35 mg/kg; Imalgene 1000; Merial S.A, Lyon, France) intravenously.

These brief explanations have been implemented in the manuscript.

REMARK 4. Line 161, 10-14 embryos per female: Which were they transferred into bilateral or unilateral oviduct?

Embryos were transferred bilaterally, and it has been clarified in the manuscript.

REMARK 5. Line 180, blood sample: Which condition was it collected in fasted or fed?

Blood samples were taken in fasted conditions.

REMARK 6. Line 180, 209: How were the sampled rabbits selected from 111 rabbits?

The animals from which the samples stemmed were selected randomly within each experimental group, ensuring that selected animal comes from a different litter (parity) in the proteomic study, while 1-2 animals were selected from each parity for the blood analysis.  It was clarified in the manuscript.

REMARK 7. Line 342: Indicate reference(s) about normal values in rabbit.

References about normal values in rabbits have been added.

Remark 8. Line 432, our previous data: Indicate reference(s).

It has been indicated, which is referred to the reference number 15.